# DARDN: A Deep-Learning Approach for CTCF Binding Sequence Classification and Oncogenic Regulatory Feature Discovery

**DOI:** 10.3390/genes15020144

**Published:** 2024-01-23

**Authors:** Hyun Jae Cho, Zhenjia Wang, Yidan Cong, Stefan Bekiranov, Aidong Zhang, Chongzhi Zang

**Affiliations:** 1Department of Computer Science, University of Virginia, Charlottesville, VA 22903, USA; hc2kc@virginia.edu; 2Center for Public Health Genomics, University of Virginia, Charlottesville, VA 22903, USA; zw5j@virginia.edu (Z.W.); congyidan@gmail.com (Y.C.); 3Department of Biochemistry and Molecular Genetics, University of Virginia, Charlottesville, VA 22903, USA; sb3de@virginia.edu

**Keywords:** CTCF, machine learning, DARDN, DeepLIFT

## Abstract

Characterization of gene regulatory mechanisms in cancer is a key task in cancer genomics. CCCTC-binding factor (CTCF), a DNA binding protein, exhibits specific binding patterns in the genome of cancer cells and has a non-canonical function to facilitate oncogenic transcription programs by cooperating with transcription factors bound at flanking distal regions. Identification of DNA sequence features from a broad genomic region that distinguish cancer-specific CTCF binding sites from regular CTCF binding sites can help find oncogenic transcription factors in a cancer type. However, the presence of long DNA sequences without localization information makes it difficult to perform conventional motif analysis. Here, we present DNAResDualNet (DARDN), a computational method that utilizes convolutional neural networks (CNNs) for predicting cancer-specific CTCF binding sites from long DNA sequences and employs DeepLIFT, a method for interpretability of deep learning models that explains the model’s output in terms of the contributions of its input features. The method is used for identifying DNA sequence features associated with cancer-specific CTCF binding. Evaluation on DNA sequences associated with CTCF binding sites in T-cell acute lymphoblastic leukemia (T-ALL) and other cancer types demonstrates DARDN’s ability in classifying DNA sequences surrounding cancer-specific CTCF binding from control constitutive CTCF binding and identifying sequence motifs for transcription factors potentially active in each specific cancer type. We identify potential oncogenic transcription factors in T-ALL, acute myeloid leukemia (AML), breast cancer (BRCA), colorectal cancer (CRC), lung adenocarcinoma (LUAD), and prostate cancer (PRAD). Our work demonstrates the power of advanced machine learning and feature discovery approach in finding biologically meaningful information from complex high-throughput sequencing data.

## 1. Introduction

Identification of cis-regulatory elements in the non-coding genome is a key task in functional and regulatory genomics research. Active cis-regulatory elements usually function as transcription factor (TF) binding sites, containing specific DNA sequence recognized and bound by the TF(s) that regulate gene expression. Most TF binding sites are located in distal enhancer regions in the genome that can be far away from their regulatory target genes. This makes it difficult to identify regulatory sequence motifs from a long DNA sequence. Distal enhancer-binding TFs interact with co-factors to execute their regulatory functions. One such example is CCCTC-binding factor (CTCF), a zinc finger protein that binds to DNA and can induce DNA looping, which anchors at topologically associating domain (TAD) boundaries and blocks cross-domain interactions [1]. Disruption of individual CTCF binding in the genome causing aberrant chromatin interaction and differential gene expression has been observed in many cellular systems [2,3]. Further, we previously showed that specific CTCF binding patterns frequently occur in many cancer types, and such aberrant CTCF binding events are induced by oncogenic TF binding at distal regions [4]. Therefore, the wide genomic regions flanking cancer-specific CTCF binding sites should contain sequence features for specific oncogenic TFs, and knowing the oncogenic factors is important for understanding the mechanisms of cancer development. In this paper, we show a deep learning-based approach for finding DNA sequence features enriched at genomic regions associated with cancer-specific CTCF binding sites (gained CTCF sites) but not at regions near cell type-conserved constitutive CTCF binding sites that frequently occur at chromatin domain boundaries in most cell types.

Conventional TF motif search methods are not feasible for this problem because the relative genomic location of the target oncogenic TF binding site relative to the cancer-specific CTCF site is unknown and can be very far, and it varies across different cancer-specific CTCF sites. Conventional DNA sequence motif search methods are not feasible also because the search space is huge and without appropriate control sequences. In fact, direct DNA sequence motif search in the gained sites was unable to yield any unambiguously enriched motifs other than CTCF itself [4]. TF-binding ChIP-seq data-based methods like BART [5] are potentially feasible but are limited to pre-identified cis-regulatory element repertoire such as the union open chromatin regions. Therefore, new computational methods need to be developed for tackling this unique but important problem with significant biological meaning.

Advanced machine learning approaches such as deep convolutional neural networks (CNN) have been popular for applications in genomics and cancer research [6,7,8,9]. In addition to solving a classification problem using deep learning, we also focus on the interpretation of the CNN model to make biologically meaningful discoveries. Specifically, with a well-trained deep neural network classifier, one can use feature discovery tools such as DeepLIFT (Deep Learning Important FeaTures) [10] to identify features from the model that imply functionally important biological insights. Compared with traditional bioinformatics algorithms, deep learning models are specifically suitable for this task because of the complexity of the problem, i.e., ultra-long DNA sequences with a huge number of features and relatively rare cancer-specific CTCF binding events.

To address this problem, we introduce DNAResDualNet (DARDN), a computational method that utilizes convolutional neural networks (CNNs) coupled with feature discovery using DeepLIFT, to identify DNA sequence features enriched in a set of long DNA sequences compared with another set of DNA sequences as control. DARDN trains a pair of deep CNN models, alongside residual connections, to enhance classification accuracy for extended input DNA sequences. It is designed to rely exclusively on DNA sequences for training without integrating other data types, making it simple to train and become versatile to be applied to similar sequence data from other biological scenarios. We demonstrate the effectiveness of DARDN in finding the simulated sequence motif from synthetic sequence data and finding the sequence motif for known oncogenic TFs such as Notch1 for T-cell acute lymphoblastic leukemia (T-ALL) data. We then apply DARDN to identify sequence motifs for potential oncogenic transcription factors for acute myeloid leukemia (AML), breast cancer (BRCA), colorectal cancer (CRC), lung adenocarcinoma (LUAD), and prostate cancer (PRAD).

## 2. Methods

### 2.1. Data and Their Representation

Genomic DNA sequences used in this study are derived from the human hg38 genome version. Foreground cancer-specific CTCF sites and constitutive CTCF sites data are obtained from our previous work [4], which identified cancer-specific CTCF binding patterns by integrative analysis of over 771 high-quality CTCF ChIP-seq datasets across a variety of different human cell types including both normal and cancer cells. For each of the six cancer types included in this work, tens to thousands of cancer-type-specific CTCF binding sites are identified in each cancer type, while 22,097 constitutive sites in the genome are conserved across cell types.

To alleviate the problem of data imbalance between the 72 T-ALL-specific CTCF sites and 22,097 constitutive CTCF sites, we perform data augmentation by reverse complementing and shifting the gained sites. Specifically, we shift the original sequences and their reverse complements to the left and to the right stochastically between 1 and 5 base pairs (bps) (Figure 1a). These data augmentation methods are commonly employed to enhance the robustness and generalizability of models, particularly in addressing challenges such as data imbalance and the limited sample sizes often encountered in biological datasets [11,12,13,14].

Each DNA sequence containing a CTCF binding site is then represented as a one-hot encoding in order to be processed by the deep neural network model. The matrix consisting of one-hot encoded DNAs is passed to a deep neural network to train the model (Figure 1b). The dimension of the matrix is 4×L, where *L* is the length of the DNA sequence. Hence, the model is flexible with DNA sequences with various length. The model produces a binary prediction of whether there is a cancer-specific CTCF binding site for each input sequence (0- or 1-labeled). We generate 10 kilo-base (kb) genomic DNA sequences centered at each T-ALL-specific CTCF site as a positive signal with label 1 and those centered at constitutive CTCF sites with label 0. We train our model to classify any 10 kb DNA sequence as either 0 or 1.

### 2.2. Evaluation Metrics

We use the Matthew’s correlation coefficient (MCC) to evaluate DARDN’s classification accuracy for predicting CTCF gained versus constitutive sites. MCC measures the correlation between the true labels and predicted labels, ranging from −1 to +1. A value of +1 indicates perfect prediction, −1 indicates total disagreement between prediction and truth, and 0 is the expected value for random guessing. MCC is calculated by dividing the covariance of the true and predicted labels by the product of their standard deviations, which is represented as
$$\mathrm{MCC}=\frac{TP\times \;TN-FP\times \;FN}{\sqrt{\left(TP+FP)(TP+FN)(TN+FP)(TN+FN\right)}},$$
where *TP* is the number of true positives, *TN* is the number of true negatives, *FP* is the number of false positives, and *FN* is the number of false negatives.

Using HOMER [15], we identify enriched motifs on CTCF gained sites in T-ALL, guided by prior findings of oncogenic motifs. We evaluate DeepLIFT’s performance by examining the ranking of the RBPJ motif, associated with an oncogene in T-ALL. For other cancer types, we rely on the literature to identify highly ranked oncogenic motifs. Our pipeline is tested for robustness using varying input sequence lengths, sampling gained sites, and sampling constitutive sites.

### 2.3. Model

Due to its exceptional capability in hierarchical feature extraction and the characteristic of being location invariant, convolutional neural networks (CNNs) have been used as a promising approach for generating informative latent feature maps as well as for various tasks using DNA sequences [6,7,8,9,16,17,18,19,20]. However, while plain CNN models are typically location invariant and can be effective for certain types of DNA sequences, we found that they are ineffective for our purposes, as shown in Table 1.

To tackle the limitations of plain CNN models, we developed DARDN (DNAResDualNet), a CNN-based model that is capable of learning of intricate relationships among distant DNA sequences even in the existence of deep convolutional layers. DARDN, as the name suggests, employs an ensemble of two CNNs with distinct initial kernel sizes for DNA sequence classification and includes residual connections to preserve complex relationships between distant DNA sequences. Having two input kernels of different sizes leverages the variability in gene sequence lengths to enable the CNNs to learn important features at different levels of granularity. In our approach, we utilized kernel sizes of 4 and 8 base pairs (bps) as hyperparameters, which are subject to optimization based on the length and type of the input DNA sequences. The selection of these particular kernel sizes was mainly driven by biological considerations. In our previous study [4], we identified RBPJ as the most enriched motif near newly acquired CTCF sites in T-ALL, leading us to use its enrichment rank as an essential metric for DARDN evaluation. The RBPJ consensus sequence, CCTGGGAA, is 8 base pairs long. Notably, its central segment, TGGGAA, which comprises 6 base pairs, shows a higher occurrence frequency, as shown in the JASPAR database (https://jaspar.elixir.no/matrix/MA1116.1/, accessed on 10 January 2024). To encompass the variability around this core segment, we chose initial kernel sizes of 4 and 8. Size 4 allows for capturing smaller patterns within the 6 base pair core, while size 8 matches the full length of the RBPJ motif, ensuring comprehensive analysis of both the core and its flanking regions. In cases where specific oncogenes are being targeted and their lengths are known, adjusting the kernel sizes to align with these lengths could yield more precise results.

We compared the performance of using one CNN model with that of an ensemble of two and three deep CNNs, and found that all converged to similar classification performance. These considerations were made based on existing work that identified improved generalization and reduced overfitting characteristics of ensemble CNN models [21]. Models with one or two CNN converged faster than the model with three networks due to having fewer parameters. On the other hand, models with two and three networks produced significantly higher logits at the output neurons than the model with a single deep convolutional network, suggesting higher confidence in their predictions. Given the faster convergence and higher confidence of models with multiple CNNs, we chose to implement our DNA sequence classifier with two networks. The two-channel CNN model yielded superior classification performance, demonstrating the benefits of our approach for accurately classifying DNA sequences. Furthermore, within each CNN model, a skip (residual) connection [22] was established from the input of the first CNN layer to the second non-linear activation to maintain important signals across sequential convolutional layers. DARDN’s architecture is visualized in Figure 1b.

Finally, the binary classification prediction of gained and constitutive CTCF sites was generated by merging the outputs from each deep CNN and passing them through a fully connected layer. To train the DARDN model, the binary cross-entropy (BCE) loss was computed between the predicted probability of each sample being a CTCF gained site $p_{i}$ and the true label $y_{i}$ for each input sequence *i*:$$\mathrm{BCELoss}=-\frac{1}{N}\sum_{i=1}^{N}\left[y_{i}\cdot \log\left(p_{i}\right)+\left(1-y_{i}\right)\cdot \log\left(1-p_{i}\right)\right],$$
where *N* is the number of input sequences. By minimizing binary cross-entropy loss, DARDN can learn to make accurate predictions on whether a CTCF site is gained or constitutive.

In this work, we demonstrate the effectiveness of our model, DARDN (DNAResDualNet), in identifying oncogenic transcription factors (TFs) associated with T cell lymphoblastic leukemia (T-ALL). Specifically, we show that DARDN is capable of accurately identifying TFs that bind to known cancer-specific CTCF binding sites in long DNA sequences.

#### Applying DeepLIFT and Motif Analysis

DARDN, as a deep neural network (DNN)-based binary classifier, has two output neurons, $o_{1}$ and $o_{2}$, outputting the logits for the input sequence being a constitutive or a gained site. When applied to $o_{1}$, DeepLIFT assigns positive contribution scores to features positively influencing the model’s classification of the input as a constitutive site and negative scores to features negatively influencing classification. Conversely, when applied to $o_{2}$, DeepLIFT assigns positive scores to features positively influencing classification of the input as a gained site and negative scores to features negatively influencing classification. Since we aimed to discover motifs positively correlated with each cancer type, we applied DeepLIFT on $o_{2}$, the neuron that produces the logits for the input being a gained site.

DeepLIFT requires a reference value, serving as a null input. It compares the differences in output values obtained by running the actual and reference inputs. This difference is allocated to each base pair through backward propagation, assigning input contribution scores. The resulting scores reflect the extent to which each base pair is responsible for the output difference from the reference. We randomly sampled 80% of the constitutive sites and used the averaged frequency at each index as the reference value. In our approach, we processed sites to be in the shape of 4×L, where 4 corresponds to the four nucleotides and *L* represents the sequence length. Once we allocated contribution scores to each base, we then performed gating to retrieve the specific nucleotide that exists in the sequence at the intended location.

After obtaining DeepLIFT scores for each gained site, we applied a sliding window of length *w* bps with a single base pair stride across the scores associated with each gained CTCF site. Each base pair was assigned a DeepLIFT score, and *w* base pair subsequences were assigned a score by averaging their individual base pair scores. The sliding window method produces a total of L−w+1 subsequences. We explored the use of 10 and 20 window sizes to determine the optimal size for identifying enriched motifs. While the resulting motif enrichments varied slightly across the different window sizes, we ultimately decided to use w=20 and 20 bp subsequences to use as a HOMER input, as this size yielded superior results. This process of subsequence selection is demonstrated in Figure 1c.

After obtaining the list of subsequences and their corresponding DeepLIFT scores, we filtered them for further analysis using motif enrichment software through two approaches. The initial step involves choosing a predetermined number of subsequences. These subsequences are selected based on having the highest positive mean contribution scores for each gained site. The second approach entails aggregating all subsequences from each gained site and then selecting a fixed number of subsequences with the highest positive mean DeepLIFT scores. The first approach may be more suitable when dominant oncogene occurrences around each CTCF gained site are consistent, while the second approach may be more advantageous when oncogene occurrences vary across CTCF gained sites. Although both approaches are viable, we chose to implement the second approach and selected 1000 subsequences with the highest positive scores. Those subsequences were fed into HOMER, with which we performed known motif analysis using the findMotifsGenome.pl module and 200 base pair search space. This resulted in the list of most highly enriched motifs, as summarized in Figure 1d.

## 3. Results

### 3.1. Performance Evaluation through Simulation

To evaluate the validity of our method and DARDN’s classification ability in detecting crucial features in DNA sequences, we conducted a preliminary test using 25,762 real DNA sequences of a length of 10,000 base pairs (bps) without CTCF binding sites. We replaced any occurrences of the RBPJ consensus sequence (CCTGGGAA) with a random 8 bp combination. Then, we inserted the RBPJ consensus sequence at ten random locations in each of the 33% of the sites (25,762×0.33×10=8500 sites). We trained DARDN on the classification of RBPJ-inserted sequences and achieved 100% accuracy on hold-out data. Subsequently, we used DeepLIFT to assign contribution scores to each base pairs (bps) in the sequence. This approach allowed us evaluation of the performance of our pipeline in accurately identifying inserted RBPJ sequences and assigning relevant scores to each bp.

We first demonstrated the assignment of DeepLIFT scores to sequences that were trained using DARDN without any RBPJ sites inserted. Since these sites were randomly selected from actual DNA sequences without any specific criteria, the scores do not exhibit any discernible pattern (Figure 2a). On the other hand, after DARDN was trained to classify RBPJ-inserted sequences, it was evident that the DeepLIFT scores at those particular locations with RBPJ insertions (indicated with red dots) were significantly greater than those at other locations (Figure 2b). This provides a clear evidence that DARDN and our score assignment work as expected.

Once the DeepLIFT score at each individual index was computed, we used a sliding window to compute the average scores of subsequences in the input sequences. Specifically, the average score at index $S_{i}$ was computed using formula $\frac{1}{w+1}\sum_{i-w/2}^{i+w/2}S_{i}$, where *w* indicates the window size. Because the RBPJ consensus sequence we inserted contains 8 bps (CCTGGGAA), for our simulation, we used w=8. In our primary experiments, we tested various values of *w* to optimize the window size for motif enrichment identification.

Figure 2c shows the peak with the highest average DeepLIFT score for each plot in Figure 2b, after re-indexing to center at zero. Lastly, in Figure 2d, we illustrate the sequence logo generated by computing the Position Weight Matrix (PWM) for the sequences that center at the highest peak at each RBPJ-inserted site. Evidently, the inserted sequence of CCTGGGAA is displayed with the highest frequency in the center, which further validates our pipeline.

### 3.2. Robustness Evaluation

To comprehensively evaluate the robustness of DARDN, we subjected it to four distinct test conditions and observed the enrichment of RBPJ, which we noted in our previous research as the most enriched motif for T-ALL [4]. The test conditions we considered were (1) modifying subsequence lengths for HOMER input: this scenario involves examining how changes in subsequence lengths influence motif rankings. This is equivalent to the window size with which we compute the running average DeepLIFT scores; (2) altering input sequence lengths: we explore how motif enrichment changes with input sequences of various lengths, specifically 5000, 10,000, and 20,000 base pairs (bps); (3) sampling background control sequences from constitutive CTCF sites: this entails studying the effect of sampling constitutive sites on motif rankings; (4) sampling foreground sequences from cancer-specific CTCF sites: we investigate the impact on motif rankings when gained sites are sampled. In our experiments, we selected 150 most statistically significant T-ALL-specific CTCF gained sites and 22,097 constitutive CTCF sites, which were subsequently centered within the sequences.

In our investigation, we explored subsequence lengths ranging from 10 to 20 base pairs (bps) and discovered that adopting a subsequence length of 20 bps consistently yielded superior rankings for RBPJ, irrespective of input sequence length (5 kbps, 10 kbps, or 20 kbps). In Figure 3a, we present the percentile rank of RBPJ across various combinations of input sequence length and subsequence length. The x-axis represents the sequence lengths of 5 kbps, 10 kbps, and 20 kbps, denoted as “short”, “moderate”, and “long”, respectively.

Using 10 bp subsequences, RBPJ achieved the 91st percentile (3 out of 32 enriched motifs) for the short input sequence length, the 97.3th percentile (7 out of 264 enriched motifs) for the moderate input sequence length, and the 96.4th percentile (9 out of 264 enriched motifs) for the long input sequence length. On the other hand, when utilizing 20 bp subsequences, RBPJ achieved the 99.7th percentile (1st out of 264 enriched motifs) for the short input sequence length, 99.2th percentile (2nd rank among 264 enriched motifs) for the moderate input sequence length, and 98.5th percentile (4th rank among 264 enriched motifs) for long input sequence length. Regarding classification accuracy, DARDN demonstrated a Matthews correlation coefficient (MCC) of 0.91 for short input sequence length, as well as 0.87 for both moderate and long input sequence lengths.

To further evaluate the robustness of DARDN, we conducted five samplings of the background constitutive sites and five separate samplings of T-ALL-specific gained sites. In each trial, we randomly selected 15,000 out of 22,097 (approximately 68%) background constitutive sites and 72 out of 150 T-ALL-specific gained sites. The performance of DARDN was individually evaluated on each set of sampled sites, and the respective results are presented in Figure 3b.

During one of the tests using sampled background constitutive sites (Run 1 in Figure 3b), we observed a decline in the average rank of RBPJ compared to our previous trials that involved the complete set of 22,097 constitutive sites. RBPJ achieved the following percentiles and rankings in the five trials: 69 percentile (10th out of 32 enriched motifs), 97.3 percentile (7th out of 264 enriched motifs), 96.2 percentile (10th out of 264 enriched motifs), 92.1 percentile (21st out of 264 enriched motifs), and 94.7 percentile (14th out of 264 enriched motifs). The corresponding MCC values for classification were 0.88, 0.92, 0.81, 0.80, and 0.84, respectively. This outcome was expected as reducing the number of background constitutive sites not only diminishes the pool of negative samples, but also can weaken the robustness of DeepLIFT reference values.

For the first sampling of foreground-specific sites, we specifically sampled 72 most significant T-ALL-specific CTCF sites, measured by the specificity and the enrichment of the occurrences. Samplings 2 through 5 involved random samplings of 72 sites from the top 150 gained sites. The MCC scores for classification and the rankings of RBPJ for these trials were notably higher than those observed in the classification involving sampled constitutive sites: 99.2nd percentile (2nd out of 264 enriched motifs), 98.9th percentile (3rd out of 264 enriched motifs), 98.1st percentile (5th out of 264 enriched motifs), 96.4th percentile (7th out of 264 enriched motifs), and 98.1st percentile (3rd out of 264 enriched motifs). Both the classification accuracy and the ranking of RBPJ reached the most significant values among the five sampling experiments of the gained sites.

In Figure 4, we present the distributions of distances between each specific CTCF site and enriched RBPJ site under the five test criteria. Figure 4a–c showcase the distance variations for different input sequence lengths of 5 k, 10 k, and 20 kbps, respectively. Figure 4d,e visualize the distances obtained by independently sampling constitutive and gained sites five times. For any input sequence length, the identified RBPJ sites may occur at any distance from foreground-specific CTCF sites, suggesting that long-range interactions exist between cooperating transcription factors and specific CTCF.

In Figure 5, we show the CTCF center to motif center distance distributions for the most enriched motifs for T-ALL, including RBPJ. As shown in Figure 5a, individual motif’s distances vary widely, while the median remains in the range from 3500 to 5000 bps away from the CTCF center. In Figure 5b, the distance distributions for the motifs in Figure 5a are plotted using 1D Gaussian smoothing. We do not observe a trend of close genomic distance between the specific CTCF binding and identified motif sites for transcription factor binding, further indicating that long-range interactions can occur at a long distance through DNA looping.

Although we also have Hi-C data pertaining to T-ALL, its resolution is inadequate for deriving meaningful insights for this project. This is because the existing resolution of Hi-C data is 10 kb, and we only search for motifs within 10 kb.

### 3.3. Application of DARDN to Diverse Cancer Types

To evaluate the adaptability of the DARDN sequence feature identification method, we applied it on five other cancer types where cancer-specific CTCF sites were identified in our previous work [4], acute myeloid leukemia (AML), breast invasive carcinoma (BRCA), colorectal cancer (CRC), lung adenocarcinoma (LUAD), and prostate adenocarcinoma (PRAD), using the moderate input sequence length of 10 kbps.

In all five cancer types, the motifs for CTCF or CTCFL (a.k.a. BORIS, a paralog of CTCF) are highly enriched near the CTCF center. As both the foreground and background sequences are centered at specific or constitutive CTCF binding sites, respectively, the enrichment of the CTCF motif indicates additional CTCF occupancy near these specific sites. This is consistent with the fact that CTCF binding exhibits a clustered pattern in the genome to maintain the higher-order chromatin structure [23].

Meanwhile, the relatively uniform distribution of the remaining motifs across the sequence length shown in the Gaussian-smoothed line plots in Figure A1, Figure A2, Figure A3, Figure A4 and Figure A5 in Appendix A indicates potential long-range interactions between CTCF and other transcription factors through looping structures. The full list of cancer-specific enriched motifs is presented in Table A1, Table A2, Table A3, Table A4 and Table A5 in Appendix A.

Overall, this pattern of enrichment and distribution of different sequence motifs surrounding cancer-specific CTCF sites suggests that the regulatory mechanisms governing gene expression are specific to each cancer type and potentially involved in the specific CTCF binding events to facilitate enhancer–promoter interactions for oncogenic transcription factors to regulate their target genes.

## 4. Conclusions

This work presents DARDN, a novel deep learning computational method using dual CNNs and DeepLIFT for long DNA sequence classification and regulatory feature discovery. DARDN accurately classifies sequences surrounding cancer-specific vs constitutive CTCF binding sites. DeepLIFT selects important subsequences for feature analysis. DARDN identifies simulated regulatory sequences and known cancer TFs like RBPJ in T-ALL. Application to AML, BRCA, CRC, LUAD, and PRAD reveals distinct TFs, implying cancer-specific regulation. DARDN provides an effective framework combining deep learning and attribution for discovering functional sequence features from long genomic data without localization, addressing a key challenge in distal regulation. Our versatile approach is broadly applicable for mining insights from diverse biological sequences. DARDN represents a powerful methodology leveraging machine learning and feature discovery for extracting biological insights from complex genomic data. It is important to acknowledge that specific training for each cancer type is essential due to the unique gene expression profiles involved.

## Figures and Tables

**Figure 1 genes-15-00144-f001:**
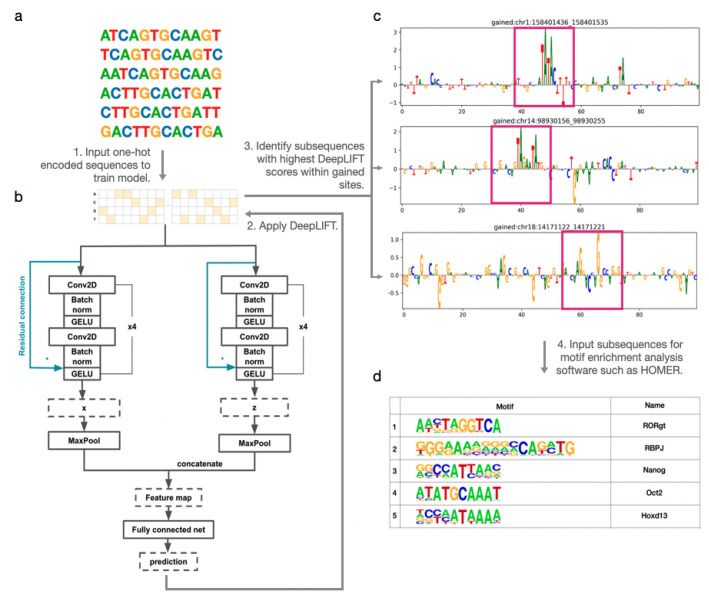
Schematic of overall computational framework design. (**a**) Data augmentation: original sequence, its left/right shifts, the reverse complement, and its left/right shifts. (**b**) DARDN Model: uses two deep convolutional networks to process a 2D one-hot encoded sequence for binary classification. (**c**) DeepLIFT is applied for sequence feature selection. Subsequences with the highest moving average DeepLIFT scores are selected for motif analysis. (**d**) HOMER motif analysis result.

**Figure 2 genes-15-00144-f002:**
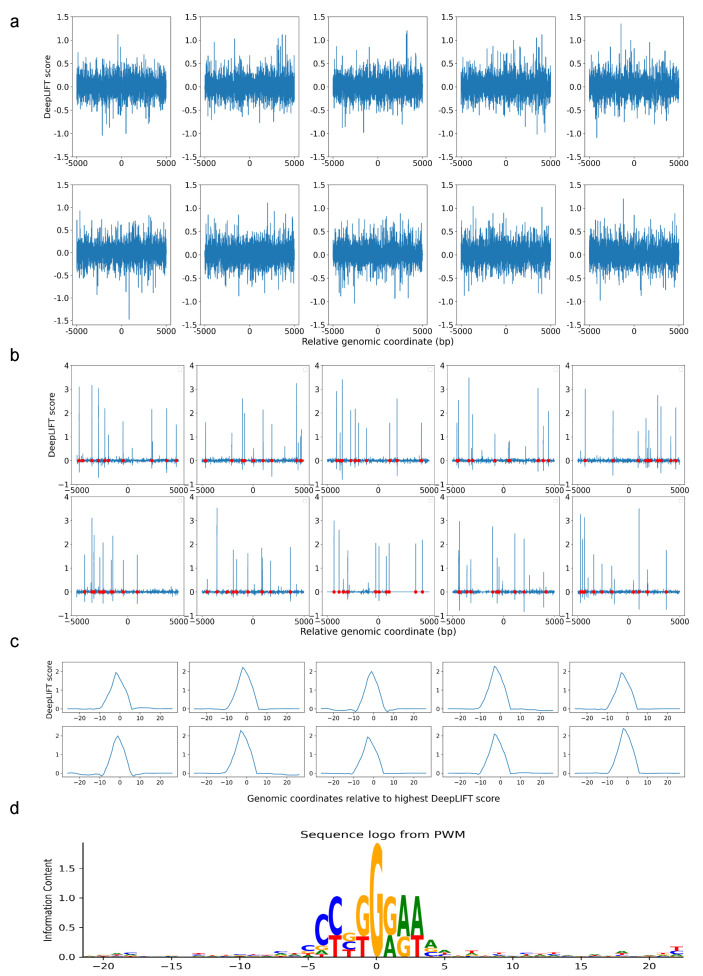
Evaluating the effectiveness of DARDN and motif discovery pipeline using simulation. (**a**) DeepLIFT scores for sites without RBPJ consensus sequence insertion. (**b**) DeepLIFT scores for sites containing 10 RBPJ consensus sequences per site, with red dots marking the locations of insertions. (**c**) Highest average scoring peak for each site in (**b**). (**d**) Sequence logo surrounding the highest peak from each RBPJ-inserted site.

**Figure 3 genes-15-00144-f003:**
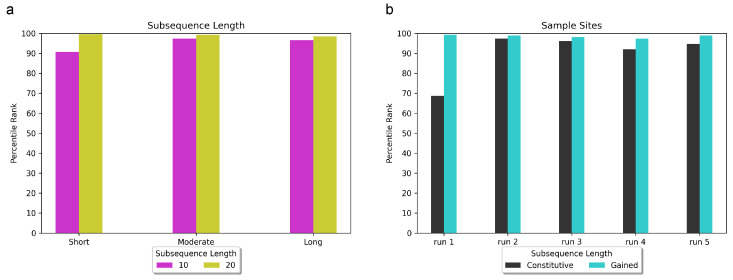
Evaluating the robustness of DARDN and our motif discovery pipeline under varying conditions, such as subsequence length, input sequence length, sampling constitutive CTCF, and sampling specific CTCF sites. Short, moderate, and long indicate input sequence length of 5 k, 10 k, and 20 kbps. (**a**) Observing the impact of subsequence length, ranging between 10 bps and 20 bps, on RBPJ rank. (**b**) Observing the impact of sampling constitutive and gained sites.

**Figure 4 genes-15-00144-f004:**
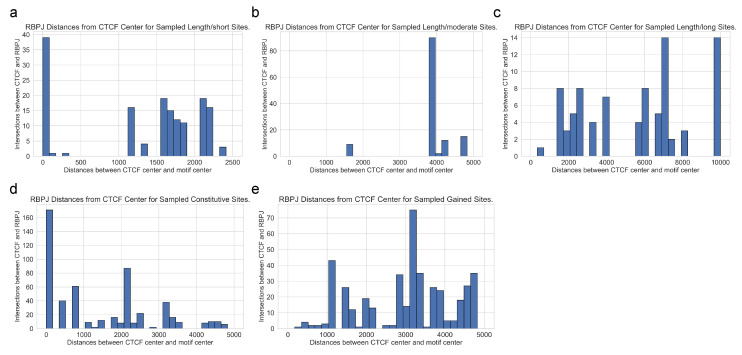
The distribution of center-to-center distances between T-ALL-specific CTCF sites and identified RBPJ sites was examined under various robustness tests. (**a**–**c**) The distributions when the input sequence lengths are 5 k, 10 k, 20 kbps respectively. (**d**) The distribution obtained by independently sampling constitutive sites 5 times. This is the aggregate distribution of the 5 sampling experiments. (**e**) The distribution obtained by independently sampling gained sites 5 times. This is the aggregate distribution of the 5 sampling experiments.

**Figure 5 genes-15-00144-f005:**
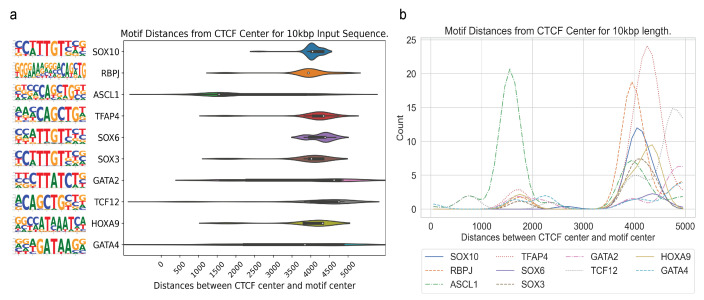
Distribution of distances for the most enriched motifs for T-ALL. (**a**) Violin plots showing the distribution of individual motif’s distance to the CTCF center. (**b**) Distance distribution for the motifs in a after applying Gaussian smoothing.

**Table 1 genes-15-00144-t001:** Evaluation on hold-out data consisting of 78 T-ALL-specific CTCF gained sites and 5129 constitutive CTCF sites of a length of 10,000 base pairs. The number of hold-out gained sites includes augmented sites. A plain CNN model is unable to make accurate classification due to class imbalance and sequence length, while our model, DARDN, can achieve significantly superior performance. MCC stands for Matthew’s Correlation Coefficient.

Model	True Positives	True Negatives	False Positives	False Negatives	MCC
CNN	47	5067	31	62	0.5
DARDN	76	5108	2	21	0.87

## Data Availability

Cancer-specific CTCF gained data used in this paper can be found here: https://zanglab.github.io/data/cancerCTCF (accessed on 17 January 2024). The code used to generate the results discussed in this paper is published here: https://github.com/berkuva/DARDN (accessed on 17 January 2024).

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
