# Peer review of "DARDN: A Deep-Learning Approach for CTCF Binding Sequence Classification and Oncogenic Regulatory Feature Discovery"

_genes, 2024, doi:10.3390/genes15020144_

Round 1

Reviewer 1 Report

Comments and Suggestions for Authors

The manuscript entitled “DARDN: A deep-learning approach for CTCF binding sequence classification and oncogenic regulatory feature discovery” presents a novel deep-learning model that combines dual CNNs and DeepLIFT for long DNA sequence clarification and discover regulatory features. This study applied the proposed DARDN model to five additional cancer types, together with the T-ALL cancer type, and demonstrated the performance of the robustness and accuracy of the model through a list of evaluations. The authors claim that their pipeline can be applied to diverse biological sequences, which sounds promising and exciting with the emergence of massive sequencing data. The manuscript is well organized. I would like to just bring up a few minor points that could further improve the clarity and consistency of the manuscript if addressed in a revision:

1.     Page 4, it stated that 4 and 8 based pair-long initial kernel sizes worked best after tuning the hyperparameters. Can any tables support this experiment's hyperparameters optimization results? The same concern applied to the final decision to use w = 20 and 20 bp subsequences as HOMER input.

2.     Page 4, it mentions the model comparison started with one, two, and three deep CNNs. Where is the “three” number coming from? Is there any published paper that can be cited as a reference?

3.     Page 4, in the main text it stated that within each CNN model, a skip residual connection is established. Thus, in figure 1b, it might be clearer to add the green box to the second CNN architecture. At first glance of the model architecture, I would think only one of the CNN models includes the residual connection, which is not correct after reading the main text.

4.     The manuscript mentioned multiple times about the gained site and constitutive CTCF site. It would be more beneficial for the readers if there were one or two sentences to introduce the definitions and differences between those two sites.

5.      Page 5, the sentence of: “The first approach is to select a fixed number of subsequences with either the highest positive mean contribution scores for each gained site.” should be rephrased.

6.     Page 9, line 314, change “a. k. a.” to “a.k.a.”, remove the space.

7.     Page 14, line 444, change the “demonstrates” to the “demonstrate.”

Comments on the Quality of English Language

Some parts of the paragraphs can be proofread and rephrased in the revised version.

Author Response

We would like to thank the reviewers and editors for taking the time to review our manuscript and their thoughtful, constructive comments. We appreciate the reviewers’ positive evaluation of the manuscript. We have responded to the reviewers’ comments with changes to the manuscript and feel that the manuscript is notably stronger.

  1.     Page 4, it stated that 4 and 8 based pair-long initial kernel sizes worked best after tuning the hyperparameters. Can any table support this experiment's hyperparameters optimization results? The same concern applied to the final decision to use w = 20 and 20 bp subsequences as HOMER input.

It is challenging to do a comprehensive analysis to show hyperparameter optimization results within the limited time frame available. Therefore, we explain the rationale behind selecting short initial kernel sizes (less than 10) below and in the manuscript [lines 139-150].

In our earlier study [1], as detailed in the manuscript, we identified RBPJ as the most enriched motif near newly acquired CTCF sites in T-ALL, leading us to use its enrichment rank as an essential metric for DARDN evaluation. The RBPJ consensus sequence, CCTGGGAA, is 8 base pairs long. Notably, its central segment, TGGGAA, which comprises 6 base pairs, shows a higher occurrence frequency, as shown in the JASPAR database (https://jaspar.elixir.no/matrix/MA1116.1/). To encompass the variability around this core segment, we chose initial kernel sizes of 4 and 8. The size 4 allows for capturing smaller patterns within the 6 base pairs core, while 8 matches the full length of the RBPJ motif, ensuring comprehensive analysis of both the core and its flanking regions.

These kernel sizes were chosen with prior knowledge about the length of the main target motif, RBPJ; hence it is important to consider that if there is prior knowledge about the length of other particular oncogenes for search, it may be more appropriate to select the kernel sizes accordingly.

As alluded in your comments and explained in the manuscript (lines 199-207), the window size (w) with which to compute the average DeepLIFT score must match the HOMER input subsequence length. We believe that Figure 3a suffices to demonstrate that using 20bp subsequences resulted in superior performance over using 10bp subsequences. Specifically, across all short (5k), medium (10k) and long (20k) bps subsequence lengths, the rank of RBPJ was higher when 20bp-long subsequences were used than when 10bp-long subsequences were used as HOMER input.

  1.     Page 4, it mentions the model comparison started with one, two, and three deep CNNs. Where is the “three” number coming from? Is there any published paper that can be cited as a reference?

The number of deep CNNs is also a hyper-parameter, where each CNN will be designed to learn slightly different attributes from the input and the final output will be an aggregation of them. We used two deep CNNs for optimal performance and computational efficiency, which is depicted in Figure 1b. Had we chosen to implement three deep CNNs, there would have been another deep CNN channel in addition to the two currently shown.

We have added a clarifying sentence on the benefits to having an ensemble of CNN models in lines 151-154 which is also included below:

We compared the performance of using one CNN model with an ensemble of two and three deep CNNs, and found that all converged to similar classification performance. These considerations were made based on previous works that have identified improved generalization and reduced overfitting characteristics of ensemble CNN models [2, 3].”

  1.     Page 4, in the main text it stated that within each CNN model, a skip residual connection is established. Thus, in figure 1b, it might be clearer to add the green box to the second CNN architecture. At first glance of the model architecture, I would think only one of the CNN models includes the residual connection, which is not correct after reading the main text.

We realized that the green box is no longer necessary and removed it from the figure to avoid any confusion, as the remaining arrows indicate the residual connection and the repeat of the two CNN layers four times. This is reflected in the updated manuscript.

  1.     The manuscript mentioned multiple times about the gained site and constitutive CTCF site. It would be more beneficial for the readers if there were one or two sentences to introduce the definitions and differences between those two sites.

We have modified the last sentence of the first paragraph in Introduction to explain gained and constitutive CTCF sites. This change is included below:

“In this paper, we show a deep learning-based approach for finding DNA sequence features enriched at genomic regions associated with cancer-specific CTCF binding sites (gained CTCF sites) but not at regions near cell type-conserved constitutive CTCF binding sites that frequently occur at chromatin domain boundaries in most cell types.”

  1.     Page 5, the sentence of: “The first approach is to select a fixed number of subsequences with either the highest positive mean contribution scores for each gained site.” should be rephrased.

The sentence is updated in the manuscript. Below is the updated version:

“The initial step involves choosing a predetermined number of subsequences. These subsequences are selected based on having the highest positive mean contribution scores for each gained site.”

  1.     Page 9, line 314, change “a. k. a.” to “a.k.a.”, remove the space.

Resolved in manuscript.

  1.     Page 14, line 444, change the “demonstrates” to the “demonstrate.”

Resolved in manuscript.

References:

[1] Fang, C., Wang, Z., Han, C. et al. Cancer-specific CTCF binding facilitates oncogenic transcriptional dysregulation. Genome Biol 21, 247 (2020). https://doi.org/10.1186/s13059-020-02152-7

[2] https://ai-scholar.tech/en/articles/deep-learning/single-ensemble

[3] Lee, YH., Won, J.H., Kim, S. et al. Advantages of deep learning with convolutional neural network in detecting disc displacement of the temporomandibular joint in magnetic resonance imaging. Sci Rep 12, 11352 (2022). https://doi.org/10.1038/s41598-022-15231-5

Reviewer 2 Report

Comments and Suggestions for Authors

In this article, the authors develop DARDN, a method based on neural networks, to detect CTCF binding regions associated to cancer. The method is tested with previously identified regions and is then applied to several cancer types. Overall, results show how this method is cable of recognizing cancer specific CTCF binding sites.

I only have the following minor comments:

1. The authors should define DeepLIFT in the abstract.

2. Proof the sentence "relative genomic location of target oncogenic TF binding site relative"

3. I wonder how this method can be translated to other TFs as the authors used their own data but do not test it with other data

4. Is there a validation for the data augmentation?

5. Can the authors correlate their results on cancer-specific enriched motifs presented in Tables A1 to A5 with Hi-C data for these cancer types? 

Author Response

We would like to thank the reviewers and editors for taking the time to review our manuscript and their thoughtful, constructive comments. We appreciate the reviewers’ positive evaluation of the manuscript. We have responded to the reviewers’ comments with changes to the manuscript and feel that the manuscript is notably stronger.

  1. The authors should define DeepLIFT in the abstract.

We have integrated a short description of DeepLIFT with an existing sentence, as shown below:

“Here we present DNAResDualNet (DARDN), a computational method that utilizes convolutional neural networks (CNNs) for predicting cancer-specific CTCF binding sites from long DNA sequences and employs DeepLIFT, a method for interpretability of deep learning models that explains the model's output in terms of the contributions of its input features [12], for identifying DNA sequence features associated with cancer-specific CTCF binding.”

  1. Proof the sentence "relative genomic location of target oncogenic TF binding site relative”

The phrase “relative genomic location of target oncogenic TF binding site relative” could not be found in the manuscript.

  1. I wonder how this method can be translated to other TFs as the authors used their own data but do not test it with other data.

Although adapting the DARDN to other TFs would require careful consideration of the specific characteristics of those TFs and retraining the model with relevant data, DARDN’s principles and methodology are applicable as the deep learning architecture for sequence analysis and feature discovery makes it versatile for similar sequence data in different biological scenarios.

  1. Is there a validation for the data augmentation?

We use reverse complementation and shifting as data augmentation methods. These methods were chosen because they both maintain the biological rationale as the original DNA sequence. For reverse complement, DNA is double-stranded, with each strand being complementary to the other. This means that for every adenine (A) in one strand, there is a thymine (T) in the opposite strand, and for every cytosine (C), there is a guanine (G) in the opposite strand, and vice versa. The biological processes that read DNA can work on either strand, so the information content is preserved in the reverse complement.

The process of shifting also does not modify the inherent structure of DNA, as it. Shifting the sequence slightly can still preserve the biological context, as these elements do not always have a fixed starting position.

Previously works have adapted both methods as primary data augmentation methods as listed below:

[1] Zhang TH, Flores M, Huang Y. ES-ARCNN: Predicting enhancer strength by using data augmentation and residual convolutional neural network. Anal Biochem. 2021 Apr 1;618:114120. doi: 10.1016/j.ab.2021.114120. Epub 2021 Jan 31. PMID: 33535061.

[2] Jakub M Bartoszewicz, Anja Seidel, Robert Rentzsch, Bernhard Y Renard, DeePaC: predicting pathogenic potential of novel DNA with reverse-complement neural networks, Bioinformatics, Volume 36, Issue 1, January 2020, Pages 81–89, https://doi.org/10.1093/bioinformatics/btz541

[3] Zhen Cao, Shihua Zhang, Simple tricks of convolutional neural network architectures improve DNA–protein binding prediction, Bioinformatics, Volume 35, Issue 11, June 2019, Pages 1837–1843, https://doi.org/10.1093/bioinformatics/bty893

[4] Avsec, Ž., Agarwal, V., Visentin, D. et al. Effective gene expression prediction from sequence by integrating long-range interactions. Nat Methods 18, 1196–1203 (2021). https://doi.org/10.1038/s41592-021-01252-x

  1. Can the authors correlate their results on cancer-specific enriched motifs presented in Tables A1 to A5 with Hi-C data for these cancer types? 

This is a great suggestion.  However, we have Hi-C data for T-ALL, but the resolution is not high enough to make any useful observations for this project. The resolution of Hi-C data is 10kb, and we only search for motifs within 10kb. There is also not enough time for processing and analyzing Hi-C data for the other cancer types. We have added the following sentence (lines 326-328) noting that this is a future direction of this project:

“In our study of T-ALL using Hi-C data, the current resolution of 10kb limits our ability to draw meaningful insights. A potential future ”

Reviewer 3 Report

Comments and Suggestions for Authors

The manuscript presents a novel approach where the authors have developed a neural network based on the residue module for the identification of cancer-specific CTCF-binding sites. Following this, the application of DeepLIFT to discern DNA sequence features linked to these sites is a notable aspect of the study. This methodology shows promise in advancing our understanding of regulatory proteins in cancer biology.

Overall, the research is conducted with a high degree of rigor and is articulately presented.

However, I would suggest that the authors elaborate on the model's ability to generalize across various types of cancer cells. It would be beneficial to clarify whether there is a need to train a distinct model for each new cancer-cell type, or if the existing model demonstrates broad applicability. This clarification would greatly enhance the understanding of the model's practical utility in diverse clinical scenarios.

Author Response

We would like to thank the reviewers and editors for taking the time to review our manuscript and their thoughtful, constructive comments. We appreciate the reviewers’ positive evaluation of the manuscript. We have responded to the reviewers’ comments with changes to the manuscript and feel that the manuscript is notably stronger. 

DARDN’s architecture was designed with versatility in mind, as the model is capable of taking in any DNA sequence. The model is trained to identify and locate genes that are differentially expressed between gained and constitutive CTCF-centered sequences. As different genes are differentially expressed for each cancer type, the model should be trained individually for each cancer type. We have included a clarifying sentence in the conclusion to reflect this, as included below:

“It is important to acknowledge that specific training for each cancer type is essential due to the unique gene regulatory program involved.” [lines 363-365]

Reviewer 4 Report

Comments and Suggestions for Authors

Cho and colleagues developed DARDN for predicting CTCF binding sequence based on deep learning, and applied it on different cancer types. Overall this is a well-performed study and well-written manuscript. I have some comments or questions as following:

1.     How does DARDN’s performance compare to other established tools for CTCF binding sequence classification?

2.     Why the authors only applied DARDN to five cancer types rather than all cancers?

Comments on the Quality of English Language

Minor editing of English language required

Author Response

We would like to thank the reviewers and editors for taking the time to review our manuscript and their thoughtful, constructive comments. We appreciate the reviewers’ positive evaluation of the manuscript. We have responded to the reviewers’ comments with changes to the manuscript and feel that the manuscript is notably stronger.  

  1.   How does DARDN’s performance compare to other established tools for CTCF binding sequence classification?

To the best of our knowledge, prior studies have not specifically focused on identifying both established and potential oncogenic transcription factors at cancer-specific CTCF-gained sites using deep learning approaches. 

  1.     Why the authors only applied DARDN to five cancer types rather than all cancers?

In the manuscript, under section “3.3 Application of DARDN to Diverse Cancer Types”, we explained why these six cancer types were chosen. We selected them (T-ALL, AML, BRCA, CRC, LUAD, PRAD) because these are the only cancer types that we have sufficient high-quality CTCF ChIP-seq data to identify cancer-specific CTCF binding sites, which are the starting input data of DARDN. The ChIP-seq data was gathered as part of our recent work [1].

References:

[1] Fang, C., Wang, Z., Han, C. et al. Cancer-specific CTCF binding facilitates oncogenic transcriptional dysregulation. Genome Biol 21, 247 (2020). https://doi.org/10.1186/s13059-020-02152-7
